# Computational design of novel Cas9 PAM-interacting domains using evolution-based modelling and structural quality assessment

**Cyril Malbranke**[1,2]*, **William Rostain**[2], **Florence Depardieu**[2], **Simona Cocco**[1], **Rémi Monasson**[1], **David Bikard**[2]

**1** Laboratory of Physics of the Ecole Normale Superieure, PSL Research, CNRS UMR 8023, Sorbonne Université, Paris, France, **2** Institut Pasteur, Université Paris Cité, CNRS UMR 6047, Synthetic Biology, Paris, France

* cyril.malbranke@phys.ens.fr

**Data Availability Statement:** Relevant data are within the manuscript and its Supporting information files. Heavy dataset and code can be

## Abstract

We present here an approach to protein design that combines (i) scarce functional information such as experimental data (ii) evolutionary information learned from a natural sequence variants and (iii) physics-grounded modeling. Using a Restricted Boltzmann Machine (RBM), we learn a sequence model of a protein family. We use semi-supervision to leverage available functional information during the RBM training. We then propose a strategy to explore the protein representation space that can be informed by external models such as an empirical force-field method (FoldX). Our approach is applied to a domain of the Cas9 protein responsible for recognition of a short DNA motif. We experimentally assess the functionality of 71 variants generated to explore a range of RBM and FoldX energies. Sequences with as many as 50 differences (20% of the protein domain) to the wild-type retained functionality. Overall, 21/71 sequences designed with our method were functional. Interestingly, 6/71 sequences showed an improved activity in comparison with the original wild-type protein sequence. These results demonstrate the interest in further exploring the synergies between machine-learning of protein sequence representations and physics grounded modeling strategies informed by structural information.

## Author summary

Proteins are essential molecules in all living organisms, with their function largely determined by their sequence. Modifying a protein's sequence to achieve a desired function remains a challenging endeavor, requiring careful consideration of factors such as the stability of the structure and interactions with molecular partners. In this study, we devised a protein design method that combines insights from experimental data, natural variation in protein sequences, and physics-based predictions. This approach provides a reliable and interpretable means of altering a protein's sequence while maintaining its functionality. We applied our technique to a domain of the Cas9 protein, a key component in the CRISPR gene editing system. Our results demonstrate the possibility of generating

found in the GitHub repository which is also linked in the manuscript https://github.com/CyrilMa/DesignCas9WithCLD.

**Funding:** SC and RM were supported by the Agence Nationale de la Recherche grant numbers ANR-17-CE30-0021 RBMPro and ANR-19-CE30-0021 Decrypted. CM is recipient of a PhD funding from AMX program, École polytechnique and benefits from financial support from the Centre de Recherche Interdisciplinary (CRI) through "École Doctorale Frontiéres de l'Innovation en Recherche et Education – Programme Bettencourt". DB, WR and FD were supported by European Research Council [677823], European Research Council [101044479], Agence Nationale de la Recherche [ANR-10-LABX-62-IBEID]. The funders had no role in study design, data collection and analysis, decision to publish, or preparation of the manuscript.

**Competing interests:** The authors have declared that no competing interests exist.

functional protein domains with over 20% of their sequence modified. These findings underscore how the integration of diverse sources of information in a unified design process enhances the quality of engineered proteins. This advancement holds promise for creating valuable protein variants for applications in drug development and various industries.

# 1 Introduction

The growing availability of large databases of protein sequence and protein structure [1, 2] is opening the way to data-driven modeling and design of proteins. When a sufficient number of sequences are available for a protein family, *i.e.*, proteins that share similar sequences, structure and functionality, it becomes possible to learn meaningful statistical features of the family of sequences. These include the position and nature of conserved residues, but also the co-variation of residues that typically occurs on evolutionary timescales when amino acids are in contact in the protein structure. Statistical models trained on multiple sequence alignments of protein families have therefore proven useful in the prediction of protein structure [3], the prediction of the effect of mutations [4], the clustering of proteins, and the design of new protein variants [5, 6].

One simple yet effective approach for modeling these statistical features relies on Direct Coupling Analysis [7–9], a set of methods that learn the couplings between the different positions of a protein, revealing contacts or interactions between amino acids. Among the different approaches to learn these couplings, the Restricted Boltzmann Machine (RBM) has the interest of providing a low dimensional continuous representation space which has been shown to be property-aware [10], and is easier to explore than the protein sequence space. However, evolution-based models often require many sequences and can have difficulties modeling the complex relationships between amino-acids, as shown by McGee et al. [11]. These sequence models also struggle to access mutations and combinations of mutations beyond those explored by evolution.

In recent years, significant efforts have been dedicated to modeling protein structures. Unlike evolutionary-based methods that rely solely on sequence information [12, 13], structural modeling of protein families incorporates physical and chemical considerations. Structural modeling approaches have been employed for various important tasks in protein design, including inferring structure from a sequence, inferring a sequence from a structure (for example backbone design [14]), assessing the quality of a structure [15–17] and studying protein interactions with ligands such as other proteins, RNA, DNA, and metals [18, 19]. Over the past two decades, successful approaches have predominantly relied on empirical force fields like Rosetta [20] or FoldX [19]. These frameworks incorporate various physicochemical properties to evaluate and optimize the tertiary structure of proteins. The recent breakthrough in protein structure prediction based on deep learning algorithms [21, 22] is now generating great improvements in computational protein design, as shown in recent works [23–25].

However, some challenges remain. Many proteins react to perturbation in their environment and contact with ligands through conformational changes to carry out their function. Design efforts relying on static structure models are unlikely to work in this context.

Beyond what can be learned from natural sequence variation and structural modeling, experimental data on protein activity and specificity is sometimes available that could inform the design of novel protein variants with properties of interest. Experimental data is however usually sparse and expensive to generate. Integrating these diverse sources of information and

modeling approaches in design strategies remains a challenge. In this work, we explore a method of semi-supervised training of an evolutionary-based model, an RBM, that enables to leverage sparse functional data to improve the statistical representation of the protein family. We built upon previous works [26, 27] that explored semi-supervision using RBM. We further propose a sampling method based on Langevin Dynamics to explore the latent representation of the RBM while constraining single or multiple features. These features can be computed using physics-grounded models of protein properties or structure prediction, allowing the incorporation of a different source of information in the design process. For the needs of this study, we chose to work with FoldX [19] a framework that uses an empirical force field built upon the chemical terms that have been deemed important for protein stability. FoldX was recently successful in various tasks, including the prediction of the effect of mutations [28–30] or protein variant design [31]. We also used AlphaFold2 to assess the quality of the proteins generated with our method.

We evaluate our strategy on the design of Cas9 variants. The Cas9 protein is an RNA guided nuclease from the prokaryotic immune system known as CRISPR-Cas9 [32]. It forms a complex with a guide RNA, which directs Cas9 to bind complementary DNA sequences and introduce a double strand break. The first step in target recognition involves binding to a small DNA sequence motif found immediately downstream of the target sequence and known as the Protospacer Adjacent Motif (PAM). This interaction is mediated by a specific domain of the Cas9 protein, the PAM interacting domain (PID). In the case of the well described Cas9 protein from Streptococcus pyogenes (SpyCas9), the PAM motif is composed of any nucleotide followed by two guanines (NGG). Previous works showed that the PID of Cas9 can be swapped between different variants while retraining Cas9's functionality ([33, 34]), suggesting that the PID is relatively independent of the rest of the protein. In this study, we thus chose to model the PID only, without integrating the rest of the enzyme into our design efforts.

Our approach enabled the generation of functional variants of the Cas9 PID that differ by more than 50 amino-acids from natural sequences. We demonstrate how it is possible to incorporate information from natural sequence variation, physical constrains and functional data all at once in the design of variants of a large complex protein for which a limited number of sequences are available. We provide evidence of how each category of information contributes to the overall success of the method.

## 2 Results

### 2.1 Chimera between natural PIDs and SpyCas9 show relative independence between the PID and the rest of the protein

Previous research demonstrated the function of Cas9 chimeras with recombinant PID, but only a handful of such chimeras exist in the literature [35, 36]. We therefore assessed the function of additional chimeras to better probe the modularity of the Cas9 protein and confirm that it is indeed reasonable to model the PID independently of the rest of the protein.

We built a plasmid carrying dCas9 under the control of an inducible pPhlF promoter and a guide RNA targeting the promoter of an operon expressing the reporter gene *mCherry* followed by the counter-selection marker *sacB* (see Fig 1**a**). With this system, cells expressing a non-functional variant of dCas9 will form red fluorescent colonies and die in the presence of sucrose, while repression of the operon by a functional dCas9 variant will yield low fluorescence and permit survival on sucrose (Fig 1**b**). The plasmid was designed so that the PID could be easily exchanged. We used this system to construct and test 10 PIDs with increasing phylogenetic distance to S. pyogenes Cas9 (Fig 1**c**).

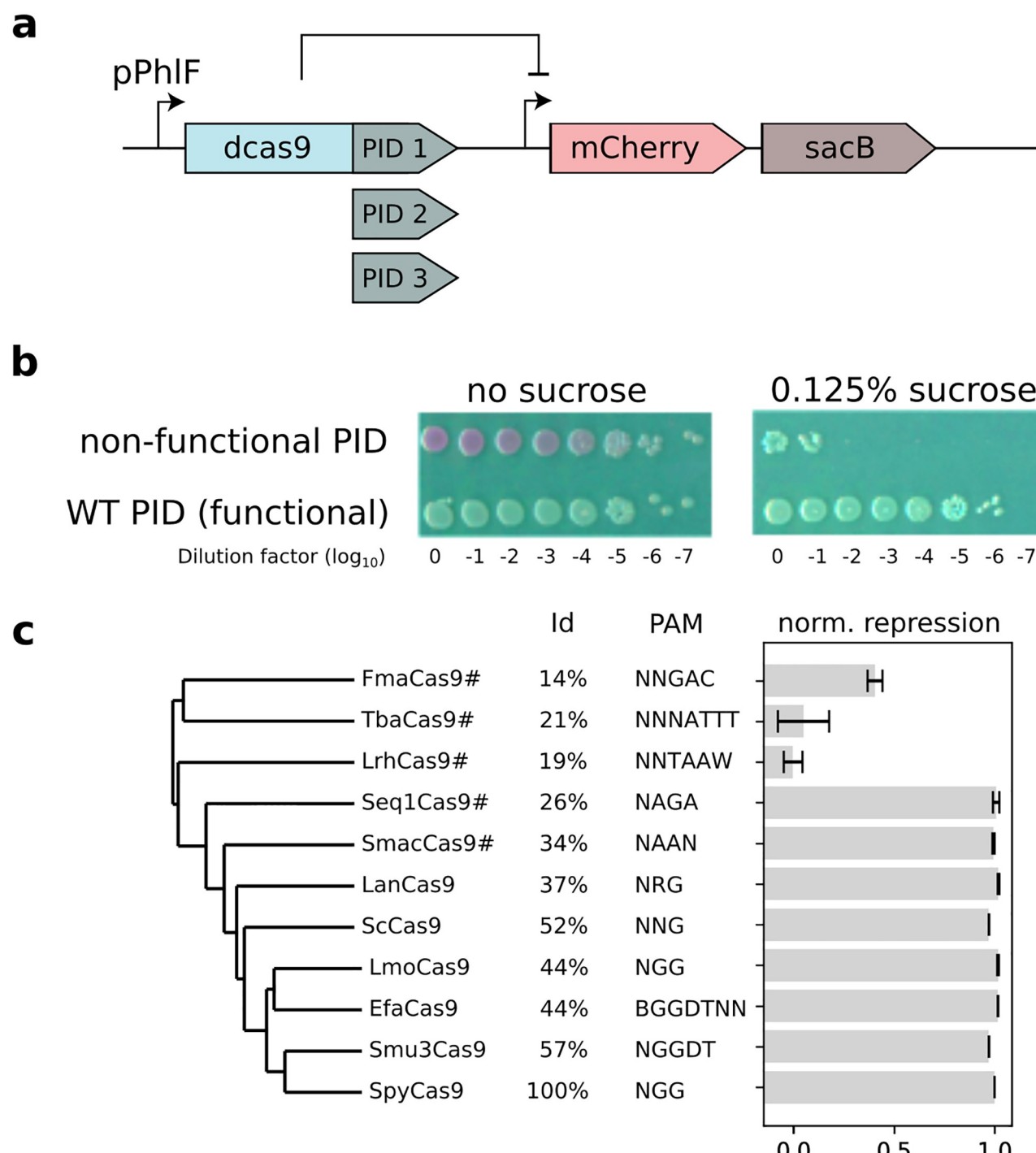

**Fig 1. Experimental validation of chimeras between SpyCas9 and the PAM-interacting domain (PID) of other natural variants. a**: Genetic circuit used to test Cas9 PID variants. The dCas9 gene is guided to silence an operon that consists of a mCherry reporter and the SacB counter-selection marker. The gene construct was designed to enable the easy exchange of PID domains (see Methods). **b**: Serial dilution and spotting of E. coli MG1655 carrying the wild-type SpydCas9 or dCas9 without a functional PID in the absence (left) or presence (right) of sucrose. **c**: The activity of dCas9 chimeras is reported as the normalized repression of the mCherry fluorescence. Chimeras were tested against targets with a PAM recognized by the PID they carry [37]. This score is normalized such that 1 corresponds to the activity of the WT and 0 to the negative control. The percentage of identity to the SpyCas9 PID is reported. We flagged with a # the PIDs which did not recognize the TGG PAM and were therefore tested with another PAM.

The activity of the chimeras are reported as the logarithm of the repression fold $f$ compared to an inactive dCas9 with no PID ($f_0$) and normalized to the WT dCas9 ($f_1$):

$$\text{normalized repression}(f) = \text{nr}(f) = \frac{\log(f/f_0)}{\log(f_1/f_0)} \tag{1}$$

An inactive sample will have a score of 0 and an active sample will have a score close to 1 if it is as active as the WT Cas9 PID.

SpyCas9 and all 7 chimeras within the group closest to SpyCas9 were functional, including those with differing PAM requirements, while in the out group consisting of FmaCas9, TbaCas9 and LrhCas9, the level of activity dropped significantly. These results show that modelling the PID on its own is a reasonable approach, but that the design of SpyCas9 chimera with a novel PID should be restrained to PID variants modelled to be similar to the ones seen to be functional here.

## 2.2 Modeling Cas9 PAM-Interacting domain with Restricted Boltzmann Machine

This section intends to present the modeling of the protein family through Restricted Boltzmann Machines (section 2.2.1) and introduce a way to better consider available functional information (section 2.2.2). In the case of Cas9 PAM-interacting domain, a major aspect of the domain's specificity is the set of PAMs that it recognizes. We show evidence that the incorporation of this functional information in the RBM improves both the ability to classify the recognized PAM sequences (section 2.2.2) and the ability of the RBM to generate high-quality sequences (section 2.4).

**2.2.1 Restricted Boltzmann Machine for protein sequence-based modeling.** RBMs are two-layer probabilistic graphical models (Fig 2) able to learn a distribution of probability over a set of inputs. Their modeling and generative capacities have driven a lot of attention these last decades [38]. Training of RBM is done through Stochastic Gradient Descent to minimize the negative log-likelihood of the training set (see Methods, section 4.2). We used Persistent Contrastive Divergence [39], which relies on approximate but efficient sampling of the model distribution.

The RBM architecture is composed of (i) a visible layer encoding in a vector $x$ an aligned protein sequence (made of amino acids and gaps), and (ii) a hidden layer composed of several continuous variables encoded in a vector $h$. These two layers are connected by the coupling

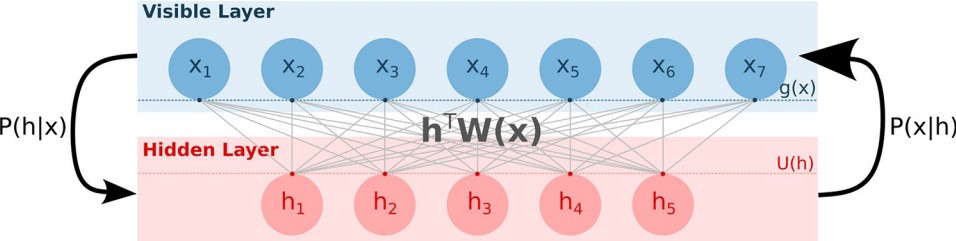

**Fig 2. Schematic representation of a standard Restricted Boltzmann Machine.** The RBM is a probabilistic graphical model with two layers: the visible layer carries protein sequences $x$ and the hidden layer encodes latent vector $h$. The graph is not oriented, allowing one (i) to sample from the visible layer to the representation layer using the conditional distribution $p(h|x)$ depending on $x$, on the weight matrix $W$ and on the potentials $U$ on the hidden units (ii) to sample from the hidden layer to the visible layer using the conditional distribution $p(x|h)$ depending on $h$, on the weight matrix $W$ and on the potentials $g$ representing the prevalence of the amino acids at each position.

**Fig 3. Semi-supervision helps the RBM learn useful representations of Cas9 PID. a**: Our Semi-Supervised Learning RBM with a one-layer classifier. This classifier takes as input the representation of the sequence in the hidden layer, and outputs the predicted PAM. **b**: Area under the ROC curve for the prediction of the PAM on the validation set. The curve is smoothed by averaging over 20 consecutive values. The shaded area shows the standard deviation over these 20 consecutive values of $\gamma$.

matrix $W^{xh}$; the potentials $g$ and $U$ act on the units on, respectively, the visible and hidden layers. Training of the parameters $\Theta_{\text{RBM}}$ can be done through the minimization of a loss function, $\mathcal{L}_{\text{RBM}}(\Theta_{\text{RBM}} = (g, U, W^{xh}))$ whose computation is detailed in methods.

Under the right choices of potentials, it is possible to ensure that conditional sampling of each visible unit from the hidden layer and back can be done following simple distributions $p(x|h)$ and $p(h|x)$ (see Methods, section 4.2 for more details).

**2.2.2 Semi-Supervised Learning Restricted Boltzmann Machines (SSL-RBM).** The standard RBM is an unsupervised model that does not require any other information than amino-acid sequences. When available, experimental data on functional protein properties can greatly enhance the ability of models to capture meaningful information (see [27, 40]). However, experimental data, hereafter referred to as labels, are usually only available for a few sequences in a protein family. We present a technique for exploiting those scarce labels using semi-supervision (see Bravi et al [27] for a semi-supervised classifier trained on top of the RBM). On top of the standard RBM architecture, we add a classifier $C$ with defining parameters $\Theta_C$, which receives as input the hidden representation of the RBM and outputs a prediction of the labels (see Fig 3a).

In SSL-RBM, the loss function used to train the parameters of the RBM, $\mathcal{L}_{\text{RBM}}(\Theta_{\text{RBM}})$, incorporates an additive contribution, $\mathcal{L}_C(\Theta_C, \Theta_{\text{RBM}})$, to fit the parameters of the classifier to the available labelled data. This approach is similar to the one proposed in [26]. The resulting loss function is

$$\mathcal{L}_{\text{SSL-RBM}}(\Theta_{\text{RBM}}, \Theta_C) = \mathcal{L}_{\text{RBM}}(\Theta_{\text{RBM}}) + \gamma \, \mathcal{L}_C(\Theta_C, \Theta_{\text{RBM}}) \tag{2}$$

We weight the classifier $\mathcal{L}_C$ and the RBM loss $\mathcal{L}_{\text{RBM}}$ to find a balance between modeling of the sequence data (minimization of the negative log-likelihood of the RBM) and performance of the classifier. If $\gamma$ is small, the model will converge to a standard RBM (since only $\mathcal{L}_{\text{RBM}}$ will be important for the minimization of the loss); if $\gamma$ is big, the model will look like a simple classifier unable to benefit from the joint modeling of unlabeled data. From the classification perspective, the RBM loss will act as a smart regularization, with unlabeled sequences structuring the RBM representations.

We evaluate the SSL-RBM method by modeling the Cas9 PID (PF16595 PFAM family, 1071 sequences). As labels, we used a dataset of PAM recognition motifs predicted for 154 PIDs using genomics data [41]. We then evaluated the predictive capabilities of our model on a held-out validation set. We used MMseqs2 to cluster the sequences with a threshold of 90% similarity, and randomly extracted a set of clusters corresponding to $\sim 15\%$ of the sequences

as our validation set. We trained 250 models with $\gamma$ ranging from $10^{-2}$ to $10^{2}$. For labelled data, each of the first 5 positions in the PAM motif was modelled independently by using as a target a binary vector encoding the recognized motifs (one unit for each nucleotide on each position, see methods in section 4.2 for more details). This enabled us to reduce the dimension of the target vector to a size that was more appropriate given the amount of available data. The RBM contained 200 hidden units and the classifier was a one-layer dense neural network. Batch Normalization was also applied to prevent over-fitting.

For each position and nucleotide of the PAM (20 units), we computed the area under the ROC curve (AUC) for the predictions of whether the nucleotide is accepted. We then reduced it to a single mean AUC by averaging over all positions and nucleotides. We show in Fig 3**b** how this mean AUC evolves over $\gamma$ for the validation set. As we see, we reach a maximum for intermediate value of $\gamma$ that outperforms a standard RBM ($\gamma = 0$), demonstrating that supervision helps the RBM separate PIDs with different recognition motifs in the latent space. Our model also outperformed a classifier trained uniquely on the labeled sequence data (high values of $\gamma$), highlighting the contribution of unlabeled data to the determination of a useful latent representation.

## 2.3 Sampling variants under constraint: Constrained Langevin Dynamics in the RBM representation space

RBMs are powerful models that create an interesting representation space in which proteins with similar properties, structure, sequence, or functionality cluster together [10]. They are also relatively light models, which makes them well suited to model small datasets. Sequence models of higher complexity are easily limited by the number of available sequences [11]. Besides, physically grounded modeling frameworks can also provide useful information (protein stability, binding affinity, etc.) without the need for large datasets. We explore here a novel strategy based on a Langevin dynamics process that enables direct sampling according to external inputs provided by a physics-grounded model (FoldX), while at the same time providing control over the region of the latent space being explored.

Sampling of Restricted Boltzmann Machine is usually done through Gibbs Sampling by alternating sampling of the visible layer from the hidden layer and the hidden layer from the visible layer. However, it was recently shown that Gibbs sampling does not guarantee that the full distribution of sequences will be sampled in a reasonable time [42], and this approach does not easily unlock the possibility of conditional sampling of sequences.

We propose another sampling strategy (see Fig 4**a**) that enables to improve one particular objective, such as the dissimilarity to the WT (to find functional sequences that are close or far from a natural sequence), while controlling a set of constraints (RBM score, structural stability...). This approach relies on two pillars.

First, we introduce an embedding of any function $f$ over the protein sequence space $x$ into the representation space $h$. For instance, $f(x)$ may be a predictor of the biochemical properties of the protein associated to $x$ or of its structural similarity with a known structure. The embedding, denoted by $\phi[f]$, estimates the average value of $f(x)$ for sequences $x$ drawn from the conditional distribution $p(x|h)$, where $h$ is a fixed representation:

$$\phi[f](h) = \mathbb{E}_{x|h}(f(x)) \tag{3}$$

The function $\phi[f]$ and its gradient are easy to estimate through sampling (see methods in section 4.3).

Introduction of the embedding $\phi$ above allows us to carry out a generalized Langevin dynamics in the representation space [43], to both optimize the objective $f$ under constraints

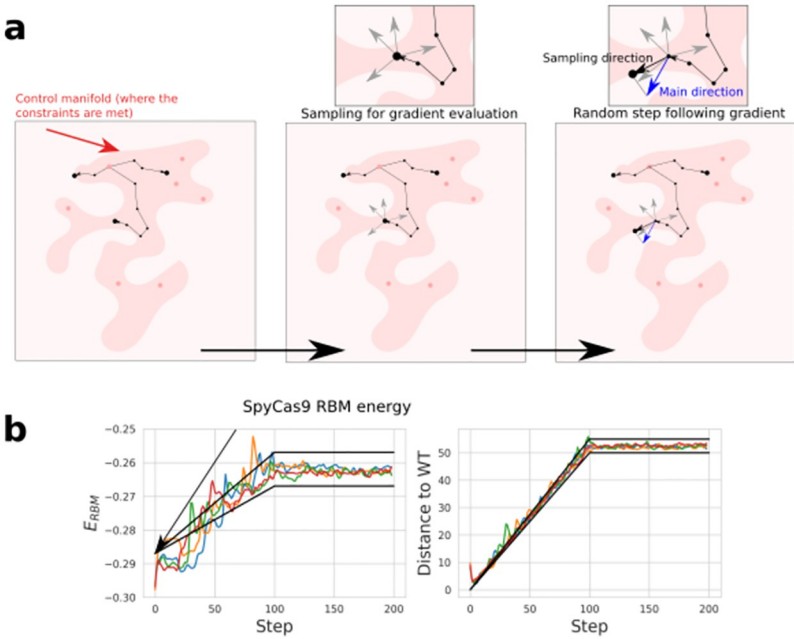

**Fig 4. Constrained Langevin Dynamics as a sampling method. a**: The Constrained Langevin Dynamics in the representation space consists of two steps. The first step is a round of sampling to evaluate the gradient of the main criterion and the gradient of the control criterion through the expectation formula (see Methods), the second is a random step following the direction drawn by these two gradients (orthogonal projection of the main direction vector regarding the control direction). A Brownian noise is added to create randomness and diversity in the samples. **b**: Typical Random Walks obtained starting from the WT SpyCas9 PID and progressively targeting an RBM energy between -0.267 and -0.257 and a Hamming distance to the WT from 50 to 55 amino acids. The black lines are representing the time-dependent target intervals along the random walk.

and create diversity. Step by step, we perform a walk in the representation space by following the gradient of $\phi[f]$ and a Brownian perturbation. We also exploit the same embedding framework to enforce a set of constraints, as the embedded functions form a support to project back the representations into the space meeting the constraints:

$$h_{t+1} = \text{Proj}\left[h_t + \alpha\, \partial_h \phi[f](h_t) + \epsilon\, \mathcal{N}(0, I)\right] \tag{4}$$

Details about this projection can be found in section 4.3.

To illustrate the control of Constrained Langevin Dynamics through RBM, we carried out random walks targeting different RBM energies and Hamming distance to the reference SpyCas9 PID. The constraints drive the sequences towards target values of similarity and energy. In Fig 4b we show two examples of random walks with 5 different chains. The sequences sampled with this approach were tested experimentally, see section 2.5.

## 2.4 Semi-supervision helps the RBM generate higher quality sequences

Before testing variants of Cas9 experimentally, we first tested the generative capacities of the SSL-RBM *in silico*. To do so, we used Constrained Langevin Dynamics to generate 120 protein sequences per RBM, with an RBM energy comparable to the one of the reference sequence (SpyCas9) and numbers of mutations ranging from 10 to 50. Sequences were generated in this manner using values of $\gamma$ ranging from $10^{-2}$ to $10^{2}$.

We then used FoldX to obtain a fast evaluation of the quality of the generated sequences (see Fig 5a). We performed the mutations from SpyCas9 WT to the sequences using FoldX

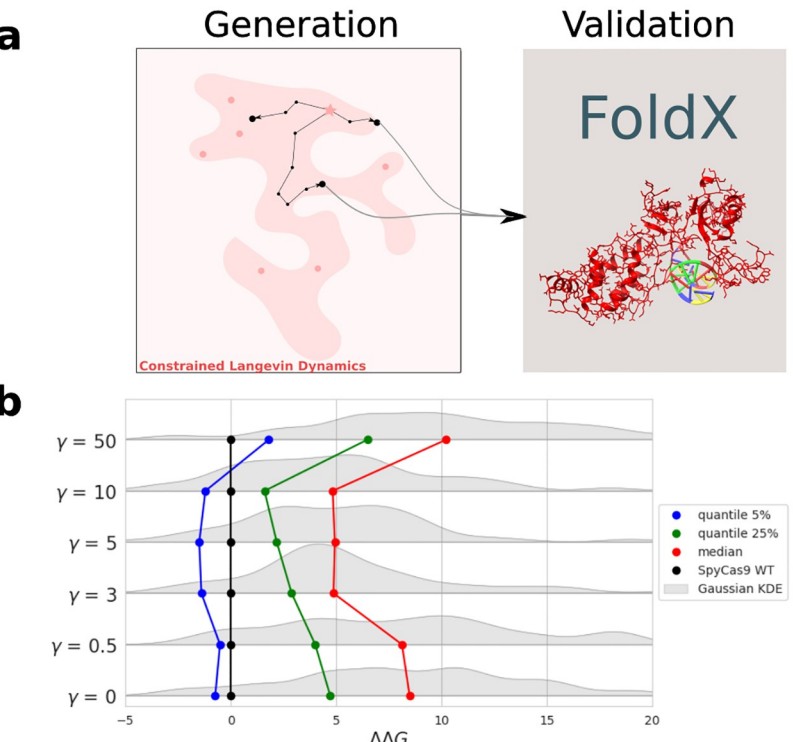

**Fig 5. Generative capacities of the SSL-RBM. a**: We tested the generative capacities of our models by generating 120 sequences with Constrained Langevin Dynamics using the trained SSL-RBM. We then use FoldX to compute the energy (displayed as ΔΔG, change in stability with respect to the wild-type) of the protein-DNA complex for the generated sequences and evaluate their quality. **b**: Distributions of FoldX energies of sequences generated with increasing values of γ. Distributions are drawn in gray using Gaussian Kernel Density Estimation, quantiles are also displayed for the different distributions. These quantiles and distributions show that overall, SSL-RBM trained with intermediate values of γ tend to generate sequences with better (lower) FoldX energies.

BuildModel and computed the ΔΔG to compare the stability of the obtained variant with that of the wild type. Variants with high ΔΔG values are more likely to carry destabilizing mutations and be non-functional.

We show the distribution of FoldX ΔΔG values obtained for different values of γ in Fig 5**b**. We observe a balanced regime where intermediate values of γ are better than in the regimes with weak (small values of γ) and with strong (high values of γ) feedbacks from the classifier. A peak is reached for values of γ around 5. This suggests that using semi-supervised training on functional data—specifically, the recognized PAM—enhances the model's capability to generate protein variants predicted to be stable by FoldX. This is promising for the creation of experimentally active sequences.

## 2.5 Design of SpyCas9 Pam-Interacting domains

**2.5.1 Sampling of the RBM-SSL using Langevin Dynamics leads to functional PAM interacting domains.**   To evaluate the efficacy of our method, we employed Langevin Dynamics to create a set of sequences with varying degrees of similarity to SpyCas9, RBM scores, and predicted levels of protein-DNA complex stability determined by FoldX energy calculations. These sequences were generated utilizing the representation space of a semi-supervised learning Restricted Boltzmann Machine (SSL-RBM) trained with a parameter value

of $\gamma = 5$. Subsequently, the designed sequences underwent experimental assessment. Detailed information regarding all the sequences, including their corresponding AlphaFold2-predicted structures, can be found in the supplementary materials (S1 File). Furthermore, computational scores and experimental validation results for these sequences are also provided (S2 File).

We adopted two different methods for sampling:

- The first method (and batch) aimed at increasing the Hamming distance to SpyCas9 WT, while constraining both the RBM log likelihood and the FoldX score within a certain interval (sequences referenced dgfx1 to dgfx29).

- The second method (and batch) constrained the RBM score and the Hamming distance to SpyCas9 within a certain interval using Langevin Dynamics, and then selected sequences with good FoldX score *a posteriori* (sequences referenced dgfx30 to dgfx75). We generated variants aiming for intervals of 5 mutations spanning from 10 to 70 mutations ([10, 15], [15, 20], . . ., [65, 70]) and intervals of RBM energy spanning from -0.21 to -0.07 ([-0.21,-0.19], . . ., [-0.09, -0.07]).

The first method is more computationally expensive than the second one, as FoldX energies need to be computed at each step of the Langevin dynamics to evaluate the gradient of this function in the representation space. In the second method, FoldX scores are only computed after sampling as a filtering step once the model is already trained. This method could be easily implemented since a large share of sampled sequences have a good FoldX score (see section 2.4).

We selected 71 sequences for experimental examination. Out of these, 29 were produced using method 1 and 42 with method 2. These sequences span a spectrum of values for metrics such as RBM energy, FoldX $\Delta\Delta G$, and AlphaFold RMSD. Our aim in selecting these sequences was to span a large range of values for these metrics in order to best evaluate which metric or combination of metric would best predict functionality. DNA sentences encoding the desired proteins were synthesized and introduced in E. coli. The ability of the generated proteins to bind a target position with a NGG PAM motif was evaluated using a mCherry reporter gene as described in section 4.5 (see Methods). The activity of dCas9 is reported as the normalized repression nr defined in Eq 1 (Fig 6). We noticed that a fair share of sequences showed a good activity score in both batches, with 17/29 sequences presenting some activity in the first batch (including 10 with an activity that matches or supersedes the WT). The second batch also displayed 8/42 active variants, including 2 with an activity similar to the WT. Notice that the second batch includes sequences designed with relaxed objectives of distance to SpyCas9 PID, RBM and FoldX energy compared to the first batch (see below). On average, the sequences in the second batch had (by design) worse RBM and FoldX energy and a larger Hamming distance to the wild-type. The lower fraction of functional variants is likely a reflection of these relaxed constraints.

In Fig 6c and 6**d**, we display the activity of the generated sequences compared to the RBM score, Hamming distance, and FoldX $\Delta\Delta G$. We see that some sequences are still active while being quite distant from the SpyCas9 sequence (50 amino acids), as long as they keep a low RBM Energy. The upper-left part of Fig 6**d** (good FoldX score, good RBM score) displayed a high proportion of active or very active sequences. We also show sequences generated by Constrained Langevin Dynamics as part of batch 2, but that were not experimentally tested. We notice that a substantial part of the generated sequences display a good FoldX score, which justifies the use of this structural score as a post-processing step rather than a constraint during Langevin Dynamics sampling in the second method.

Finally, we can also stress that our method produced functional variants with mutations at the DNA binding interface, as well as mutations at positions identified as critical for protein

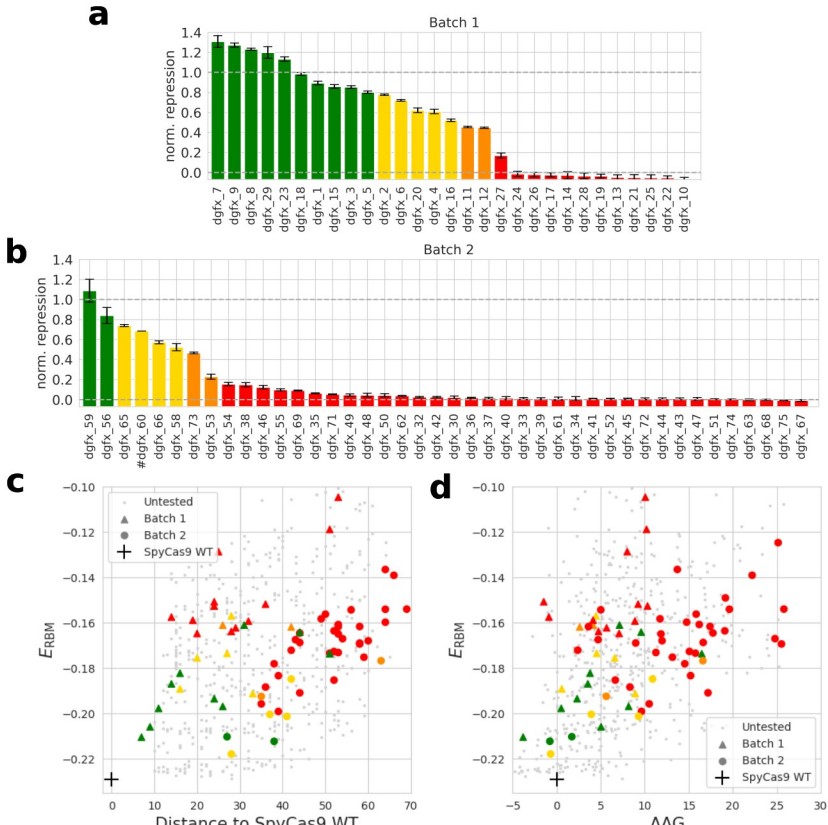

**Fig 6. Experimental evaluation of generated sequences and relationship with Hamming distance, RBM and FoldX scores. a** and **b**: The activity of the dCas9 chimera carrying generated PIDs was measured using the mCherry fluorescence assay. Arbitrary thresholds were used to classify proteins as very functional (green, nr > 0.8), functional (yellow, 0.5 < nr < 0.8), marginally functional (orange, 0.2 < nr < 0.5), not functional (red, nr < 0.2). Bar plot **a** represents the sequence in the first batch of tested sequences, and bar plot **b** represents the sequences in the second batch of tested sequences. Error bars are plotted using 2 $\sigma$, where $\sigma$ in the standard deviation between independent measurements (see Methods) (# dgfx_60 large standard deviation obtained from experimental measurement was removed from the figure for clarity) **c**: Scatter plot of generated protein sequences as a function of the RBM energy and Hamming distance of the sequence to that of the WT. Gray dots are untested sequences generated through Constrained Langevin Dynamics. We found very active sequences with up to 50 differences to the WT and any known natural sequence **d**: Scatter plot of generated protein sequences as a function of the RBM energy and FoldX $\Delta\Delta G$.

stability, with these mutations being only slightly less frequent than mutations at other positions (details can be found in supplementary (Fig C and D in S3 File)). This indicates that the method can perform mutation on critical area of the protein while retaining the activity of the variants.

**2.5.2 Semi-supervision of the RBM leads to better prediction of Cas9 activity.** We then investigated whether our experimental data would corroborate the *in silico* results obtained in section 2.4 that semi-supervision improved the ability of the RBM to generate good sequences. To that end, we computed the correlation between the RBM score ($E_{\text{RBM}}$) and the experimentally determined Cas9 activity, using models trained with different values of $\gamma$ (the strength of the classifier during training). We can see (Fig 7) that the correlation between the RBM score and Cas9 activity improves when the unsupervised and supervised losses are balanced. Interestingly, the optimal value of $\gamma$ in this situation is slightly higher than the optimal value for the classification task, even though a higher $\gamma$ gives more strength to the classifier.

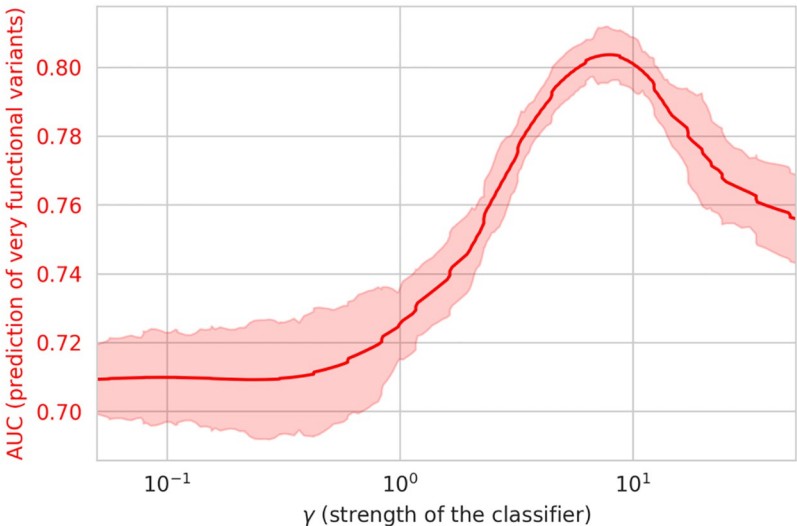

**Fig 7. Correlation between SSL-RBM energy and experimental activity.** The correlation between the experimental activity and the SSL-RBM score is plotted as a function of the classifier strength ($\gamma$) (red curve). The curve is smoothed by averaging over 20 consecutive values. The shaded area shows the standard deviation over these 20 consecutive values of $\gamma$.

Overall, these findings offer additional evidence that semi-supervision enhances both the RBM's capacity to separate Cas9 PIDs based on their PAM recognition sequences and its ability to identify functional sequences. This further confirms that the inclusion of extra information through this approach generally enhances the model's performance, even when the supervision did not relate to predicting activity, but rather specificity.

**2.5.3 A combination of sequence and structural information best predicts sequence activity.** While our RBM score is on its own a good predictor of Cas9 activity, we see that the FoldX score provides complementary information and that the combination of both features can be advantageously used to identify active Cas9 variants.

We further attempted to compare the information obtained by FoldX with the more recent protein structure prediction tool AlphaFold2 [21]. For each experimentally tested variant, we predicted a model of the structure (using ColabFold [44]). This model was then aligned with the AlphaFold2-predicted model for the wild-type protein using the TMscore program [45]. From this alignment, we derived the root-mean-square deviation (RMSD) between the models. RMSD measures the root of the mean squared distance between the backbone atoms of the two protein models. We also collected from AlphaFold2 the average pLDDT score for each model, pLDDT is a per-residue confidence score indicating the reliability of the predicted position for each amino acid in the protein structure, which in the past has been shown to correlate well with variant activity. This score can be found in supplementary (Fig B in S3 File). We found that the RMSD (AUC = 0.810) performed better than the pLDDT (AUC = 0.668) to predict the activity of experimentally measured variants.

Last, we also compared the experimental data with the classifier score for predicting how likely a PID sequence is to recognize a NGG motif.

On Fig 8a, we show the experimental activity of the enzymes given their structural scores, RBM score and classifier score. We also computed the Area Under the ROC Curve (AUROC) for the detection of active samples.

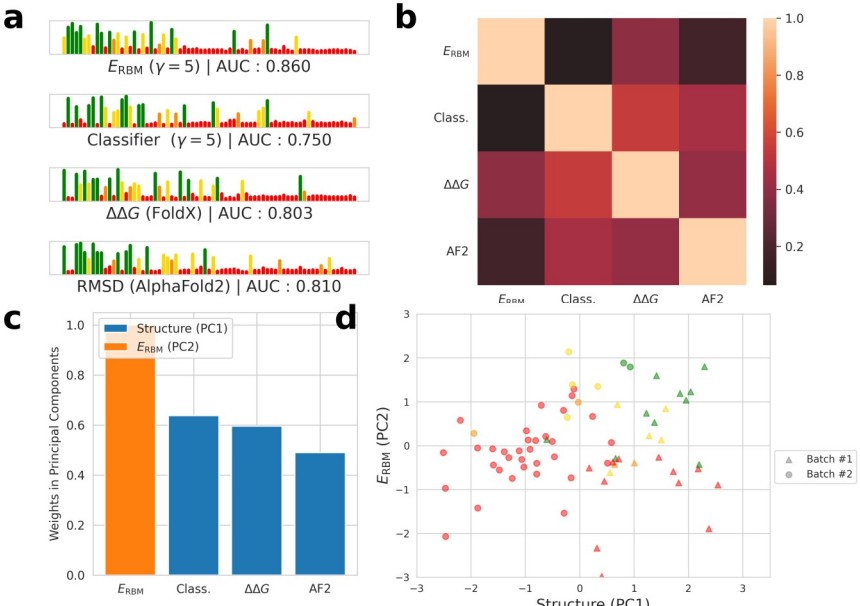

**Fig 8. RBM energy and structure scores predict sequence activity. a**: Sequences ordered by their score according to each of the metrics, the height of the bars and their color corresponds to the activity measured experimentally (color code identical to that of Fig 6). The RBM score displayed the best Spearman correlation, with AlphaFold2 showing also very good performance in detecting functional samples (AUROC curve for detecting sample with activity over 0.5). **b**: Correlation between the different scores. **c**: Three components emerge from a sparse PCA: a structure component (combination of AlphaFold2, FoldX and classifier scores), the RBM energy and the classifier score. **d**: Experimental results displayed along the two principal components inferred by the sparse PCA. The Cas9 PID variants are colored according to their activity, as in Fig 6.

All scores were able to identify functional samples (activity over 0.5) with relatively high accuracy, with the RBM displaying the best results, followed by the AlphaFold2 RMSD score and $\Delta\Delta G$ from FoldX. Interestingly, models seemed to capture complementary information, as can be seen in the failure of structural scores to spot inactive samples which are correctly predicted as active by the RBM (red dots on the bottom-right part of Fig 8d) and vice versa (red dots on the top-left part of Fig 8d).

We noticed that the classifier, AlphaFold2 and FoldX scores were highly correlated (Fig 8b). A Sparse PCA performed on the vectors of the different scores associated to each sequence emphasized these two main components: the first one encompassing the structural scores from FoldX and AlphaFold2 as well as the classifier, and the second one the RBM score alone (Fig 8c and 8d). This confirms our observation that the structural scores are strongly correlated, while the SSL-RBM brings another view of the data. We also trained Logistic Regressions with cross validation using the different scores and computed the Area Under the Curve for their ability to predict functional variants. While the RBM energy alone had an AUC of 0.86 (see on Fig 8a), combining the RBM energy with the classifier score (AUC = 0.920), the structural scores (AUC = 0.918), or both (AUC = 0.929) improved discriminative performance, as displayed in supplementary materials (Fig A in S3 File).

The classifier score (pseudo-likelihood score for the recognition of the NGG motif) correlates well with the activity and is complementary to the RBM energy, see Fig 8b. Let us, however, stress that we focused in this work on the NGG PAM only. The reason to do so is two-fold. First, the NGG motif is well represented in our initial labelled data. Second, this motif is recognized by our reference enzyme (SpyCas9) as well as by the large majority of close natural

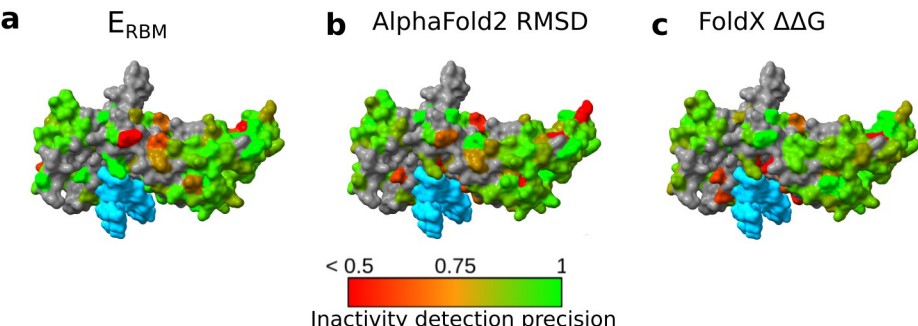

**Fig 9. Strengths and Weaknesses of Scoring Methods.** Each panel represents the precision in detecting inactive variants using different scores: RBM energy (**a**), AlphaFold2 RMSD (**b**), and FoldX's ΔΔ*G* (**c**). These metrics are plotted against the mutation's location within the protein sequence and structure. DNA regions are highlighted in blue. Green signifies regions where the deleterious impact of mutations is accurately predicted. Conversely, red indicates regions where predictions miss the mark. Areas with insufficient data (less than five variant with a mutation at this position) are marked in gray. Notably, the RBM energy offers precise predictions at the DNA-binding interface, in contrast to AlphaFold2 RMSD and ΔΔ*G*, which tend to be less accurate at these sites.

variants present in our dataset, as seen in the phylogenetic tree provided in supplementary (S5 File). The lack of examples of close variants with different PAM sequences in our dataset makes it difficult for our method to propose such variants.

**2.5.4 Strengths and weaknesses of the RBM and structural scores.** To better understand the respective strengths and weaknesses of the RBM and structural scores, we used the experimentally tested sequences to investigate whether specific positions in the structure were better modeled by the RBM or by AlphaFold2. For each residue on the protein sequences, we gathered all the inactive sequences (normalized repression below 0.2) with a mutation at this position. We then checked whether the RBM energy, the AlphaFold2 RMSD or FoldX stability score could be used to correctly predict that the variants were inactive. We computed the ratio between the number of sequences correctly predicted as inactive by each of the scores and the total number of inactive sequences. On Fig 9 we display this accuracy for the RBM energy (Fig 9a), AlphaFold2 (Fig 9b) and FoldX ΔΔ*G* (Fig 9c).

One of the key observations is that AlphaFold2 and FoldX performed more poorly than the RBM on residues that interact with DNA (represented in sky blue on Fig 9b and 9c). This can likely be explained by the fact that (i) AlphaFold2 was not trained to consider interactions with other molecular partners, such as DNA and (ii) The FoldX stability score is not suited to analyze the functionality of a binding site where the binding affinity to a specific DNA sequence is more important than stability considerations.

## 3 Discussions

Structural biology has been a field of major breakthroughs these last years with the increasing success of Deep Learning models for secondary structure prediction [46, 47], ligand or protein binding site prediction [48, 49] and of course tertiary structure prediction with frameworks such as AlphaFold2 [21], RoseTTAFold [22] or ESM Fold [50]. Older frameworks like Rosetta [20], Amber [51] or FoldX [19] have been developed around an empirical force field to encompass molecular and atomistic interactions. These frameworks rely on strong physicochemical priors designed to tackle the general lack of structural data. Even today, while hundreds of millions of protein sequences are available in public databases, only tens of thousands of structures are available since their determination is complex and expensive. Force fields can

evaluate the compatibility between an amino acid sequence and a given structure, and are generally more robust to over-fitting than data-based models.

Machine learning methods have also been applied to the modeling of proteins based on their primary sequence information alone. Such efforts range from simple models, like the RBM employed here, trained on single protein families, to large language models trained on hundreds of million of diverse protein sequences with impressive successes in various tasks [12, 13, 52–54]. Interestingly, these models typically provide a property-aware latent representation of the protein that can be leveraged together with the generative capabilities of the models for various design tasks [55–57]. We investigate here a strategy to combine these evolutionary models together with physically grounded models for protein design, in line with previous works to leverage the strength of both approaches [58].

We propose the use of Langevin Dynamics as a method enabling the exploration of a latent representation informed by external optimization goals, such as physics-grounded models of the protein properties. Here, we demonstrated this by constraining the latent space exploration using FoldX energies. Alternatively, physics grounded optimization goals can be used as a post-filtering step on proteins generated by the evolutionary model. Our method yielded functional Cas9-PAM Interacting domain variants differing by more than 50 amino-acids from any known natural sequence. The data generated highlighted how evolution-based modeling and structural modeling capture different information, and how the combination of both can advantageously be used to identify functional variants with high confidence. AlphaFold2 and FoldX can predict the protein structure but miss some information about the functionality of the sequence that can be inferred directly from the first and second order statistics of the Multiple Sequence Alignment. This observation could drive further theoretical works to better understand and circumvent the blind spots of structural modeling. One possibility is that RBMs capture information related to the protein biological function, such as its ability to undergo conformational changes or to interact with other molecular substrates, that are reflected in the sequence statistics of the protein family. Such information might, however, be missed by AlphaFold2 or FoldX if it is not directly relevant to the stability of the protein fold.

In this work, we also demonstrated how scarce functional data about the protein family can be used to improve the model using semi-supervision. We used data on the PAM binding sequence of the Cas9 PID available for 14% of our dataset. The use of this data during RBM training improved both the ability of our model to predict PAM sequences based on the PID sequence, and more interestingly contributed to improving the accuracy of our model in predicting the activity of Cas9 variants. Surprisingly, our model generated Cas9 variants that were more active than SpyCas9 for the task of gene silencing, even if this was not an explicit goal of our design. A possible explanation is that the constraints we placed on FoldX energies led to the generation of proteins that bind the target DNA more stably and are therefore more active in the gene silencing task. A future direction of this work will be to employ our models to generate Cas9 variants with modified PAM specificity while maintaining wild-type level efficiency or even improving on it.

Altogether, our results highlight the possibility to further improve our protein modeling capabilities by incorporating, structural, evolutionary but also functional information into our models and protein design process.

## 4 Material & methods

### 4.1 Dataset construction

**4.1.1 Data collection.** The dataset used in this study is the PFAM family PF16595 which contained 1102 sequences at the time of data collection, including the PID of SpyCas9.

Supervision data for the SSL-RBM consisted in the PAM recognition sequence of 154 PID variants obtained from Vink et al. [41]. This dataset contains PAM sequences inferred from the alignment of regions adjacent to target sequences of natural CRISPR-Cas systems.

**4.1.2 Data engineering.**   To build a validation set, sequences of the PF16595 PFAM family were clustered using MMSEQS2 with a 90% similarity threshold [59]. Clusters were randomly extracted one by one to build a validation set of around 10% of the total set, ensuring that sequences in both sets labelled or unlabeled were not too similar. Sequences were also weighted based on the size of the cluster they belong to, in order to avoid assigning excessive importance to data that is too similar. Concretely, let $x$ be a sequence and $\mathcal{C}(x)$ its cluster built through MMSEQS2. Over a mini-batch, each sequence is weighted according to $\frac{1}{\mathrm{card}\ \mathcal{C}(x)}$ for the computation of the loss, following the protocol proposed in Tubiana et al. [10]. This ensures that all clusters have the same importance over a full iteration of the data, which reduces the biases that could emerge from the composition of the training set.

## 4.2 Restricted Boltzmann Machine

**4.2.1 Architecture.**   The RBM has a 2-layers architecture, the visible layer has 736 visible units (length of the Multiple Sequence Alignment), encoded by Multinoulli variables with 21 states (20 amino acids + gap). The hidden layer includes 200 Gaussian units. Each of the hidden units is connected to each of the visible units, ensuring conditional sampling of each visible unit from the hidden layer and back. The weights are initialized with Xavier initialization [60], following the distribution $\mathcal{U}\left(-\frac{1}{\sqrt{qn_x}}, \frac{1}{\sqrt{qn_x}}\right)$ where $n_x = 736$ is the length of the sequences and $q = 21$ the number of states per unit.

We define $x = (x_i)_i$ an amino-acid sequence of length $n_x$. This first layer of the RBM is defined by $g(x)$ where $g$ is a linear potential that indirectly represents the prevalence of different amino acids at each position. The hidden units $h_\mu$, $\mu = 1, \ldots, 200$ are subject to the potentials $U_\mu(h_\mu)$. These two layers are linked by the coupling matrix $W_{xh}$ of size $n_x \times n_h$ such that:

$$P(x, h) = \frac{1}{Z} \exp\left[\sum_i g_i(x_i) + \sum_\mu U_\mu(h_\mu) + \sum_{i,\mu} h_\mu W_{i\mu}^{xh}(x_i)\right] \tag{5}$$

where $g_i$ and $W$ are trainable and $U_\mu(h_\mu) = \frac{1}{2}h_\mu^2$ is fixed.

The conditional probability of hidden unit $\mu$ reads

$$P(h_\mu|x) = \frac{1}{\sqrt{2\pi}} \exp\left[-\frac{1}{2}\left(h_\mu - \sum_i W_{i\mu}^{xh}(x_i)\right)^2\right],$$

which can be simulated with a Gaussian Law $\mathcal{N}\left(\left(\sum_i W_{i\mu}^{xh}(x_i)\right)_\mu, I\right)$. Similarly, the conditional probability of visible unit $i$ reads

$$P(x_i|h) = \mathrm{softmax}\left[g_i(x_i) + \sum_\mu h_\mu W_{i\mu}^{xh}(x_i)\right],$$

Which can be simulated with a multinomial law.

The SSL-RBM (see Fig 10) has the same architecture, except that we add a classifier to the previous architecture. The classifier is a one-layer neural network taking as inputs the hidden configuration (after sampling according to the potentials) and after a batch normalization layer. This ensures that more diverse input will feed the classifier (data augmentation), which enhances the quality of training.

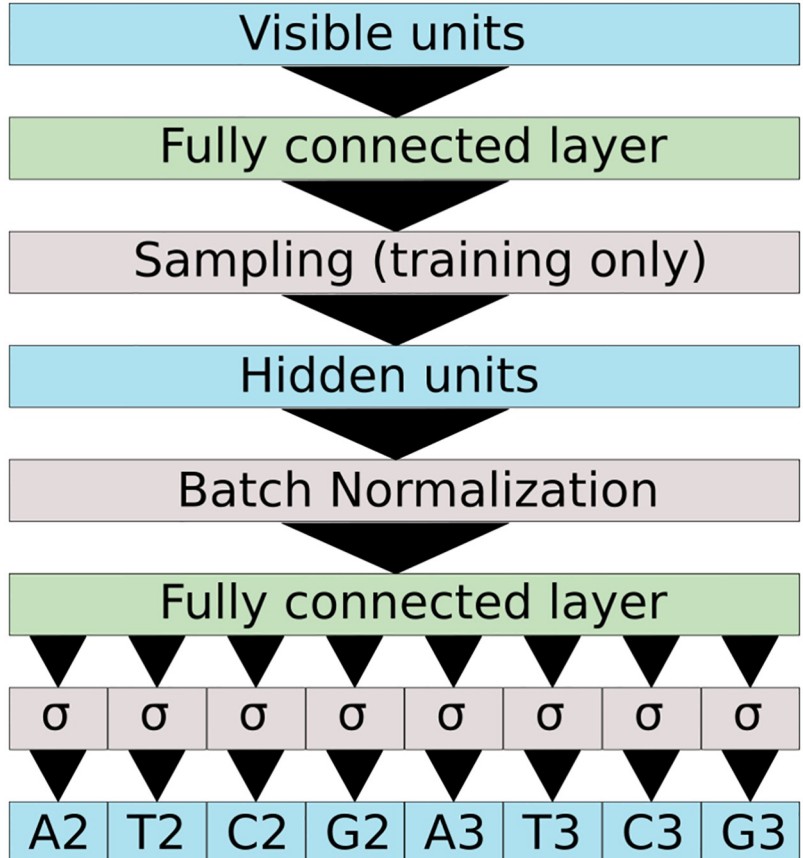

**Fig 10. Architecture of the SSL-RBM classifier.** During training the classifier takes as input the hidden layer after sampling (equivalent to a noisy representation of the input sequence) while during prediction it uses the hidden representation. The hidden representation is then batch normalized and fed into a fully connected layer. The network attempts to predict if the PID recognizes each nucleotide at each position and in the PAM.

The classifier (Fig 10) was trained to predict PAM recognized by the PAM-Interacting domain in input. We encoded the first $n$ (here $n = 5$) positions of the motifs and the four nucleotides in a $\{0, 1\}^{n \times 4} = (A_1, T_1, C_1, G_1, \ldots, T_n, C_n, G_n)$ vector with $X_k = 1$ if nucleotide $X$ is accepted in position $k$. For example, a NGGNN motif will be encoded by (1,1,1,1,0,0,0,1,0,0,0,1,1,1,1,1,1,1,1,1).

**4.2.2 Training the RBM.** The training of RBM is done through the maximization of the likelihood of the training set $T$. Let $\Theta_{\mathrm{RBM}}$ be the set trainable parameters we want to optimize. In the general case we have $\Theta_{\mathrm{RBM}} = (g, U, W)$. We denote by:

$$p(x|\Theta_{\mathrm{RBM}}) = \log\left[\frac{1}{Z(\Theta_{\mathrm{RBM}})} \exp\left[-E_{\mathrm{RBM}}(x|\Theta_{\mathrm{RBM}})\right]\right] = -E_{\mathrm{RBM}}(x|\Theta_{\mathrm{RBM}}) - \log Z(\Theta_{\mathrm{RBM}}) \quad (6)$$

the log-likelihood of a sequence $x$ given the set of parameters $\Theta_{\mathrm{RBM}}$. We perform maximum likelihood estimation on the training set to optimize the parameters.

$$\Theta_{\mathrm{RBM}}^{*} = \arg\max_{\Theta_{\mathrm{RBM}}} \sum_{x \in T} p(x|\Theta_{\mathrm{RBM}}) \quad (7)$$

We then denote the loss function for training the RBM through maximum likelihood estimation to be:

$$\mathcal{L}_{\text{RBM}}(\Theta_{\text{RBM}}) = \mathbb{E}_{x \sim T}[-\log P(x|\Theta_{\text{RBM}})] \tag{8}$$

One way to do this is to perform Gradient Descent. We have to estimate:

$$\mathbb{E}_{x \sim T}\left[\frac{\partial p(x|\Theta_{\text{RBM}})}{\partial \Theta_{\text{RBM}}}\right] = -\mathbb{E}_{x \sim T}\left[\frac{\partial E_{\text{RBM}}(x|\Theta_{\text{RBM}})}{\partial \Theta_{\text{RBM}}}\right] - \frac{\partial \log Z(\Theta_{\text{RBM}})}{\partial \Theta_{\text{RBM}}} \tag{9}$$

It is easy to approximate the first term with mini batches of samples from the training set $T$. The other term, however, is trickier:

$$\frac{\partial \log Z(\Theta_{\text{RBM}})}{\partial \Theta_{\text{RBM}}} = \frac{1}{Z(\Theta_{\text{RBM}})} \frac{\partial}{\partial \Theta_{\text{RBM}}} \sum_x e^{-E_{\text{RBM}}(x|\Theta_{\text{RBM}})} \tag{10}$$

$$= -\sum_x \frac{\partial E_{\text{RBM}}(x|\Theta_{\text{RBM}})}{\partial \Theta_{\text{RBM}}} P(x|\Theta_{\text{RBM}}) \tag{11}$$

where $P_{\Theta_{\text{RBM}}}$ the distribution of the RBM model, denoted by $M$. We then have

$$\mathbb{E}_{x \sim T}\left[\frac{\partial p(x|\Theta_{\text{RBM}})}{\partial \Theta_{\text{RBM}}}\right] = \mathbb{E}_{x \sim M}\left[\frac{\partial E_{\text{RBM}}(x|\Theta_{\text{RBM}})}{\partial \Theta_{\text{RBM}}}\right] - \mathbb{E}_{x \sim T}\left[\frac{\partial E_{\text{RBM}}(x|\Theta_{\text{RBM}})}{\partial \Theta_{\text{RBM}}}\right] \tag{12}$$

While sampling from the training set is easy, sampling from the model distribution is harder. For this, we mainly rely on **Persistent Contrastive Divergence** [39]. We implemented our own RBM in PyTorch (available on https://github.com/CyrilMa/torchpgm-standalone) running on GPU, and we used the AdamW [61] optimizer as it yielded a more robust training.

The number of neurons in the hidden layer (200) was fixed arbitrarily at an early stage by a quick exploration of the number of Gaussian units. We built a training/validation set on PF16595, using MMSEQS2 to ensure that sequences in the validation set were different enough from sequences in the training set, and we optimized the number of hidden units so to avoid overfitting (when the number of hidden units was too high).

**4.2.3 Training the SSL-RBM.** Training the Semi-supervised Learning RBM requires the use of an additional training set $T_{\text{sup}}$ of data containing a set of sequences $x$ labelled by $y$. We add a term to the loss function defined previously so that we can train both the parameters of the RBM $\Theta_{\text{RBM}}$ and the parameters of the classifier $\Theta_C$:

$$\mathcal{L}_{\text{SSL-RBM}}(\Theta_{\text{RBM}}, \Theta_C) = \mathcal{L}_{\text{RBM}}(\Theta_{\text{RBM}}) + \gamma \mathcal{L}_C(\Theta_C, \Theta_{\text{RBM}}) \tag{13}$$

where $\gamma$ is a parameter that weights the relative contribution of each term of the loss function.

In the case of a classification task, we want to minimize a cross entropy term over the training set $T_{\text{sup}}$ composed of labelled data:

$$\mathcal{L}_C(\Theta_C, \Theta_{\text{RBM}}) = \mathbb{E}_{x \sim T_{sup}, h \sim (h|x, \Theta_{\text{RBM}})}\left[\sum_i y_i \log (C_{\Theta_C}(h))_i\right] \tag{14}$$

where $y = (y_i)_i$ is a vector encoding for a label (specifically the PAM) and $C_{\Theta_C}(h)$ is the output vector of the classifier which takes in input stochastic representation of $x$. Estimation of this loss function is done using mini batches of labelled protein sequences $x$, with $h$ sampled from the distribution of $x|h$. We point out that the loss of the classifier $\mathcal{L}_C$ depends on the parameters of the RBM.

### 4.3 Sampling RBM through Constrained Langevin Dynamics

**4.3.1 Function embedding in RBM representation space.** Let $f$ be an evaluation function that on the visible space, that is, on a sequence $x$. We are interested in the optimization of this function in the hidden space. To achieve this, we consider $\phi[f](h) = \mathbb{E}_{x|h}[f(x)]$ and its gradient $\partial\phi[f](h) = \vec{\nabla}_h\phi[f](h)$. We can write

$$\vec{\nabla}_h\mathbb{E}_{x|h}(f(x)) = \vec{\nabla}_h\sum_x P(x|h)f(x) = \sum_x f(x)P(x|h)\vec{\nabla}_h\log P(x|h) \tag{15}$$

and,

$$\vec{\nabla}_h\log P(x|h) = Wx - \sum_x Wx\,\mathbb{P}(x|h) = Wx - \mathbb{E}_{x|h}(Wx) \tag{16}$$

We conclude that:

$$\vec{\nabla}_h\mathbb{E}_{x|h}(f(x)) = \sum_x f(x)P(x|h)\vec{\nabla}_h\log P(x|h) \tag{17}$$

$$= \sum_x f(x)(Wx - \mathbb{E}_{x|h}(Wx))\mathbb{P}(x|h) \tag{18}$$

$$= \mathbb{E}_{x|h}(f(x)Wx) - \mathbb{E}_{x|h}(f(x))\mathbb{E}_{x|h}(Wx) \tag{19}$$

We then have a simple way to get estimates of $\phi[f](h) = \mathbb{E}_{x|h}[f(x)]$ and of its gradient $\vec{\nabla}_h\phi[f](h)$ by sampling the conditional probability distribution $x|h$, which has been shown to be easy above. We denote by $\hat{\phi}[f](h)$ and $\hat{\partial}\phi[f](h)$ these estimators.

**4.3.2 Improving an objective through Langevin Dynamics (LD) in representation space.** Let $f$ be a criterion in the visible space, then by considering $\phi[f]$ and $\partial\phi[f]$ defined in section 4.3.1, we can build a directed random walk as follows:

**Initialization**: Let $x_0$ be the starting sequence. We initialize our random walk by $h_0 = \arg\max_h \mathbb{P}(h|x)$, which, for quadratic potential $U$, is simply $h_0 = Wx_0$.

**Recurrence**: Let $h_t$ be our position in the representation space at time $t$, then we update

$$h_{t+1} = h_t + \gamma(\hat{\partial}\phi[f](h_t) + \epsilon\mathcal{N}(0, I))\ , \tag{20}$$

where $\gamma$ and $\epsilon$ are two parameters to be chosen wisely and where $\mathcal{N}(0, I)$ is a Brownian noise.

We then have in fact two sources of randomness in our random walk. The first one is the Brownian noise, controlled by $\epsilon$, and the second one is $(\hat{\partial}\phi[f](h_t) - \partial\phi[f](h_t))$, which depends on the variance of the estimator (and thus can be lowered by raising the number of samples if needed).

This procedure, however, is problematic because the norm of $h$ tends to increase to values not encountered for representation of samples in the training set. In other words, we would explore a region of the representation space that does not correspond to our protein family data.

To address this problem, we introduced a regularization term, and now look for the maximum of $\phi'[f](h) = \phi[f](h) - \frac{1}{2}\lambda\|h\|_2^2$. This modifies the update rule in Eq (20) as follows:

$$h_{t+1} = h_t + \gamma(\hat{\partial}\phi[f](h_t) + \epsilon\mathcal{N}(0, I) - \lambda h_t)\ ,$$

with $\lambda$ to be appropriately chosen. We then rewrite this equation as

$$h_{t+1} = h_t + \gamma \ , \mathcal{C}_{\epsilon,\lambda}[f](h_t)$$

### 4.3.3 Controlling constraints through Constrained Langevin Dynamic (CLD).

Let $g$ a function (constraint) in the visible space that we want to keep fixed, say, to zero. We consider $\phi[g]$ and $\partial\phi[g]$ as defined in section 4.3.1. We propose to update $h_{t+1}$ according to

$$h_{t+1} = h_t + \gamma[\mathcal{C}_{\epsilon,\lambda}[f](h_t) + b_t \, \hat{\partial}\phi[g](h_t)] \tag{21}$$

where $b_t$ is chosen to ensure $\partial\phi[g](h_{t+1}) = 0$. To do so, using Taylor formula, we can approximate $\phi(h_{t+1})$:

$$\phi[g](h_{t+1}) \approx \hat{\phi}[g](h_t) + \partial\phi[g](h_t) \cdot (h_{t+1} - h_t) \tag{22}$$

$$\approx \hat{\phi}[g](h_t) + \gamma \, \partial\phi[g](h_t) \cdot [\mathcal{C}_{\epsilon,\lambda}[f](h_t) + b_t \, \hat{\partial}\phi[g](h_t)] \tag{23}$$

Imposing that $\phi[g](h_{t+1}) = 0$ we obtain:

$$b_t = \frac{1}{\|\partial\phi[g](h_t)\|_2^2} \left[ \frac{1}{\gamma}\hat{\phi}[g](h_t) - \partial\phi[g](h_t) \cdot \mathcal{C}_{\epsilon,\lambda}[f](h_t) \right] \tag{24}$$

Finally, we can also choose dynamical criterion with function changing over time.

### 4.3.4 Objective and constraints function.

Here we detail the function that we used as our criteria for sampling

- **Difference to reference sequence** (first batch)
  For the first batch we had difficulties finding sequences distant from the reference sequence due to strong constraints. This then became our main criterion to maximize

$$f(x) = \sum_i \mathbf{1}(x_i = x_i^0) \tag{25}$$

As for the control criteria, here are the details:

- **RBM score** (first and second batch)
  Another criterion we control is the RBM score $P(x)$ defined in section 4.2 which estimates how well the interactions between the residues in the protein are reproduced. Starting from the RBM energy of SpyCas9, $e_0$ we also target different intervals comprised between $e_{\min}$ and $e_{\max}$

$$f(x) = \left( (e_{\max} - e_{\min}) + \frac{t}{100}e_{\min} - E_{\mathrm{RBM}}(x) \right)^+ + \left( E_{\mathrm{RBM}}(x) - \frac{t}{100}e_{\min} \right)^+ \tag{26}$$

- **Function prediction** (first batch)
  We also use as criterion the output of complex function in the visible space. We then choose to maintain the FoldX scores $f$ previously defined below a certain threshold through a control criterion of the form:

$$f(x) = (f_a(x) - c)^+ \tag{27}$$

where $c$ is the maximal value we allow $f_a(x)$ to have. This option was dropped for the second batch.

- **Difference to reference sequence** (second batch)
  Difference to reference sequence was controlled starting with a linear transition from the SpyCas9 sequence $x^0$ to the expected distance, followed by a phase where this distance is kept fixed. For $t < 100$ and a final similarity in an interval comprised between $c_{\min}$ and $c_{\max}$ we have:

$$f(x) = \left( (c_{\max} - c_{\min}) + \frac{t}{100} c_{\min} - \sum_i \mathbf{1}(x_i = x_i^0) \right)^+ + \left( \sum_i \mathbf{1}(x_i = x_i^0) - \frac{t}{100} c_{\min} \right)^+ \quad (28)$$

## 4.4 Computation of FoldX ΔΔG

We collected the PDB structure of Cas9 PAM-Interacting domain (PDBid: 5y36) in complex with DNA and used the RepairPDB function from FoldX to optimize the quality of the model.

**4.4.1 Quick computation for first method design.** As explained in section 4.5, the first method we propose requires the quick evaluation of the stability of multiple variants with few differences to the variant of the previous step. To achieve this, we mutated the residues and used the Optimize function from FoldX to fix the side-chains. We then used the Stability function to evaluate the stability of the variant.

**4.4.2 Precise computation a posteriori.** All variants that were experimentally tested as well as additional untested variants had their stability re-evaluated using the FoldX framework. For each variant, we build the list of mutations to the wild-type and used the BuildModel function to evaluate stability. You can see these stability estimates on Fig 6 for the experimentally tested variants and on Fig 5 for the untested ones.

## 4.5 Experimental validation

**4.5.1 Cloning and bacterial growth conditions.** PID variants were cloned into plasmid pWR5 depicted in the supplementary information (Fig A in S4 File). pWR5 was constructed based on plasmid pFR56 [62] (GenBank: MT412099.1), a p15a plasmid with a chloramphenicol resistance gene carrying dCas9 expressed under the control of the DAPG inducible pPhlF promoter and a guide RNA under the control of a constitutive promoter. To obtain plasmid pWR5, the PID domain of dCas9 was replaced by a ccdB gene framed by BsaI restriction sites, enabling to easily swap in new PID sequences. In addition, an operon consisting of the mCherry gene followed by the sacB gene under the control of a constitutive promoter was added to the plasmid downstream of the dCas9 gene. Finally, the plasmid contains a guide RNA targeting the promoter of the mCherry-sacB operon. To test the functionality of PID with different PAM recognition sequences, we also constructed plasmid pWR6 which carries BsmBI restriction sites enabling to modify the sequence of the mCherry-sacB promoter and swap in target sites with different PAM motifs. Plasmids were cloned in *E. coli* DH10B, then introduced and characterized in *E. coli* MG1655.

**4.5.2 Sucrose resistance and fluorescence assays.** Resistance to sucrose was assessed by inoculating 1 mL cultures with individual colonies in 96 deep-well plates, refreshed 1:200 with inducer for 6h. 5 $\mu$L of serial dilutions were drop-plated onto square LB-agar plates supplemented with 25 $\mu$g/ml chloramphenicol, 50 $\mu$M DAPG and sucrose (0.125% or 0.25%) as appropriate, then grown overnight and imaged (Fig C and D in S4 File). Fluorescence was

measured by growing individual colonies in 96 deep-well plates overnight, refreshing 1:200-fold, growing 5 h then measuring the $OD_{600}$ (9 nm bandwidth) and red fluorescence (ex. 580/9 nm, em. 620/20 nm) of 200 $\mu$L of culture in a black 96 well plate (Corning costar 3603) using a Tecan M200 plate reader. Unless noted, all PID variants were tested with a guide targeting a sequenced flanked by a standard NGG PAM (TGGGTTT). Fusions targeting non-NGG PAMs were tested with the same gRNA, but the PAM sequence adjacent to the gRNA target was modified to: TAGACTT (Seq1Cas9 and FmaCas9 PIDs), TAAATTT (TbaCas9 and SmacCas9 PIDs) or TTTAAAT (LrhCas9 PID). Sequences other than the PID and the 7-nt PAM were identical in all PID tests.

**4.5.3 Phylogenetic tree of Cas9 PID chimeras.** Cas9 sequences were obtained from [37], and whole Cas9 sequences or PID domains only were aligned using MUSCLE then maximum-likelihood phylogenetic trees were built using MEGA11 [63]. The tree can be found in supplementary (Fig B in S4 File).

## Supporting information

**S1 File. Experimentally tested sequences.** This file contains the experimentally tested sequences in a zip format. It includes Fasta files, AlphaFold Prediction with PDB file, pLDDT, and more.
(ZIP)

**S2 File. Sequences with experimental and computational scores.** This Excel file contains sequences along with their corresponding experimental and computational scores.
(XLSX)

**S3 File. Additional plots on assessments of computational Score.** This PDF file contains additional plots that assess the computational score in combination to predict the activity of variants.
(PDF)

**S4 File. Experimental measurements of the activity through both mCherry and sacB.** This PDF file contains experimental measurements of the activity through both mCherry and sacB.
(PDF)

**S5 File. Phylogenetic tree of all variants selected from Vink et al.** This PNG file contains the phylogenetic tree of all variants selected from the study by Vink et al. [41].
(TIF)

## Author Contributions

**Conceptualization:** Cyril Malbranke, Simona Cocco, Rémi Monasson, David Bikard.

**Data curation:** Cyril Malbranke.

**Investigation:** Cyril Malbranke, William Rostain, Simona Cocco, Rémi Monasson, David Bikard.

**Methodology:** Cyril Malbranke, William Rostain, Florence Depardieu, Simona Cocco, Rémi Monasson, David Bikard.

**Project administration:** Simona Cocco, Rémi Monasson, David Bikard.

**Resources:** Simona Cocco, Rémi Monasson, David Bikard.

**Software:** Cyril Malbranke.

**Supervision:** Simona Cocco, Rémi Monasson, David Bikard.

**Visualization:** Cyril Malbranke, William Rostain.

**Writing – original draft:** Cyril Malbranke, William Rostain.

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
