## [Decision Letter · Decision Letter 0]

6 Jul 2023

Dear M. Malbranke,

Thank you very much for submitting your manuscript "Computational design of novel Cas9 PAM-interacting domains using evolution-based modelling and structural quality assessment" for consideration at PLOS Computational Biology.

As with all papers reviewed by the journal, your manuscript was reviewed by members of the editorial board and by two independent reviewers. In light of the reviews (below this email), we would like to invite the resubmission of a significantly-revised version that takes into account the reviewers' comments.

We cannot make any decision about publication until we have seen the revised manuscript and your response to the reviewers' comments. Your revised manuscript is also likely to be sent to reviewers for further evaluation.

Sincerely,

Joanna Slusky, Ph.D.

Academic Editor

PLOS Computational Biology

William Noble

Section Editor

PLOS Computational Biology

Reviewer's Responses to Questions

**Comments to the Authors:**

Reviewer #1: Malbranke et al. present a framework for integrating sequence information, structural modeling scores and experimental data into protein design. The authors use a Restricted Boltzmann Machine (RBM) to learn a continuous representation space for a protein family, in this particular study the Cas9 PAM-interacting domain (PID). For incorporating functional data, a semi-supervised learning (SSL) RBM was trained, aiming to improve the statistical representation of the protein family learned by the RBM. For designing new sequences of the Cas9 PID, the latent space learned by the RBM is explored using Langevin dynamics guided by constraints such as RBM score and FoldX force field energy. 71 designed variants were experimentally tested; 21 were functional and 6 had an improved activity compared to the wildtype protein. Some designs that retained function had up to 50 sequence changes compared to wildtype. Interestingly, sequences designed with both constraints, RBM score and FoldX energy, showed the highest activity on average.

The paper is well written and the results and methods are well explained. The study nicely demonstrates the combination of protein sequence machine learning models with structure-based modeling for protein design.

I have only a few minor comments / questions.

Minor comments:

Can the authors comment on how the number of neurons in the hidden layer (200 Gaussian units) was chosen?

Can the authors clarify which FoldX score (ΔΔG ?) and which AlphaFold score (pLDDT or pTM ?) were used for sequence function prediction and/or comparison with the activity of tested variants?

Page 3, line 1: “PID from SpCas9” should be “PID from SpyCas9”

Page 4, Fig 1 caption: “gene contruct” should be “gene construct”

Page 4, Fig1 caption: “exchance” should be “exchange”

Page 11, line 8: “bottom-right part” should be “upper-left part” I think.

Reviewer #2: review attached as pdf

**Have the authors made all data and (if applicable) computational code underlying the findings in their manuscript fully available?**

Reviewer #1: Yes

Reviewer #2: Yes

PLOS authors have the option to publish the peer review history of their article (what does this mean?). If published, this will include your full peer review and any attached files.

Reviewer #1: No

Reviewer #2: No
---

## [Editor Report · Decision Letter 1]

19 Oct 2023

Dear M. Malbranke,

We are pleased to inform you that your manuscript 'Computational design of novel Cas9 PAM-interacting domains using evolution-based modelling and structural quality assessment' has been provisionally accepted for publication in PLOS Computational Biology.

Best regards,

Joanna Slusky, Ph.D.

Academic Editor

PLOS Computational Biology

William Noble

Section Editor

PLOS Computational Biology

---

## [Editor Report · Acceptance letter]

13 Nov 2023

PCOMPBIOL-D-23-00836R1 

Computational design of novel Cas9 PAM-interacting domains using evolution-based modelling and structural quality assessment

Dear Dr Malbranke,

I am pleased to inform you that your manuscript has been formally accepted for publication in PLOS Computational Biology. Your manuscript is now with our production department and you will be notified of the publication date in due course.

With kind regards,

Anita Estes
